# A hitchhiker guide to manta rays: Patterns of association between *Mobula alfredi, M. birostris*, their symbionts, and other fishes in the Maldives

Aimee E. Nicholson-Jack[1,2◉]*, Joanna L. Harris[1,3◉], Kirsty Ballard[1‡], Katy M. E. Turner[2‡], Guy M. W. Stevens[1◉]

**1** The Manta Trust, Dorset, United Kingdom, **2** School of Veterinary Science, University of Bristol, Bristol, United Kingdom, **3** School of Biological and Marine Sciences, University of Plymouth, Plymouth, United Kingdom

◉ These authors contributed equally to this work.
‡ These authors also contributed equally to this work.
* aimee.nicholson-jack@mantatrust.org

**Data Availability Statement:** All sightings data used in this study is available from http://

## Abstract

Despite being among the largest and most charismatic species in the marine environment, considerable gaps remain in our understanding of the behavioural ecology of manta rays (*Mobula alfredi, M. birostris*). Manta rays are often sighted in association with an array of smaller hitchhiker fish species, which utilise their hosts as a sanctuary for shelter, protection, and the sustenance they provide. Species interactions, rather than the species at the individual level, determine the ecological processes that drive community dynamics, support biodiversity and ecosystem health. Thus, understanding the associations within marine communities is critical to implementing effective conservation and management. However, the underlying patterns between manta rays, their symbionts, and other hitchhiker species remain elusive. Here, we explore the spatial and temporal variation in hitchhiker presence with *M. alfredi* and *M. birostris* throughout the Maldives and investigate the factors which may influence association using generalised linear mixed effects models (GLMM). For the first time, associations between *M. alfredi* and *M. birostris* with hitchhiker species other than those belonging to the family Echeneidae are described. A variation in the species of hitchhiker associated with *M. alfredi* and *M. birostris* was identified, with sharksucker remora (*Echeneis naucrates*) and giant remora (*Remora remora*) being the most common, respectively. Spatiotemporal variation in the presence of manta rays was identified as a driver for the occurrence of ephemeral hitchhiker associations. Near-term pregnant female *M. alfredi*, and *M. alfredi* at cleaning stations, had the highest likelihood of an association with adult *E. naucrates*. Juvenile *E. naucrates* were more likely to be associated with juvenile *M. alfredi*, and a seasonal trend in *E. naucrates* host association was identified. *Remora* were most likely to be present with female *M. birostris*, and a mean number of 1.5 ± 0.5 *R. remora* were observed per *M. birostris*. It is hoped these initial findings will serve as the basis for future work into the complex relationships between manta rays and their hitchhikers.

idthemanta-intg.eu-west-2.elasticbeanstalk.com/home#!/home.

**Funding:** Author who received the award: G M W Stevens. Funder: Save Our Seas Foundation https://saveourseas.com/. Funding for open access publication fee if accepted. The funders had no role in study design, data collection and analysis, decision to publish, or preparation of the manuscript.

**Competing interests:** The authors have declared that no competing interests exist.

## Introduction

Symbiosis, when considered biologically, describes a physically close and long-term association between two different species [1–3]. Symbiotic interactions are common in marine ecosystems and are fundamental in regulating the distribution, abundance, and diversity of many taxa [4, 5]. Algae-coral, anemonefish, and cleaner-client mutualisms all provide traditional examples [6–9], where at least one of the interacting species is obligately dependant on the association for all, or part, of its life-history [1, 10]. While some interactions have resulted in significant behavioural adaptions and coevolution, the competitive life of a marine species can encourage short-term and opportunistic associations in order to gain food or protection [11–15]. For example, species of the family Carangidae have been observed to associate with scalloped hammerheads (*Sphyrna lewini*) to get closer to prey items, and when following cownose rays (*Rhinoptera bonasus*), cobia (*Rachycentron canadum*) have been observed to occupy a position above their host to forage on prey rejected by the rays [16]. Pilot fish (*Naucrates doctor*) are known to commonly associate with large-bodied vertebrates such as sharks, rays and turtles [17], presumably for protection from predation [14].

Species engage in associations that vary in all degrees of intimacy, ranging from obligate to facultative, mutualistic to parasitic, and long-lived to ephemeral [1, 10, 11, 18]. These interactions, rather than the species at the individual level, determine the ecological processes that drive community dynamics, support biodiversity and ecosystem health [19]. Thus, species should not be considered in isolation, and understanding the associations within marine communities is critical to implementing effective conservation and management [5, 18, 20]. Studies that incorporate habitat-specific interactions provide an opportunity to unveil population-wide and long-term patterns into the spatial and behavioural ecology of marine fauna [20–23]. However, our understanding of marine symbionts remains limited due to the logistical challenges associated with studying complex associations in mobile organisms over large spatial scales [21, 24, 25].

Manta rays (*Mobula alfredi*, *M. birostris*) are large, filter-feeding batoid rays, with a pelagic existence. *Mobula birostris* has a circumglobal distribution, while *M. alfredi* has a semi-circumglobal distribution; both in tropical and subtropical waters [26–28]. They are characteristically slow to mature, have low fecundity, and exhibit migratory and aggregatory behaviours, rendering them significantly vulnerable to exploitation [26, 27]. Consequently, and because of targeted and bycatch fisheries, *M. alfredi* and *M. birostris* are classified as Vulnerable and Endangered on the IUCN's Red List of Threatened Species, respectively [29–31]. Therefore, successful conservation of these species depends upon bridging knowledge gaps in their biology and ecology [28, 32].

The Maldives archipelago supports globally significant populations of both species of manta ray [27]. Here, coastal reef manta rays (*M. alfredi*) are commonly found throughout the archipelago, where they migrate east to west through the atolls during the transition into the Northeast (NE) Monsoon (December–March), and west to east during the onset of the Southwest (SW) Monsoon (April–November) [33]. These biannual seasonal migrations determine changes in aggregation site use, as well as the predominant behavioural activities exhibited by the highly philopatric *M. alfredi* [33, 34]. Unlike the local patterns of residency exhibited by *M. alfredi*, oceanic manta rays (*M. birostris*) are only sighted with regularity in the Maldives' southernmost atolls of Addu and Fuvahmulah, and only for a few months each year (March–April) during the transition from the NE to the SW Monsoon [27, 35]. These southern Maldives sites are in close proximity to deep-water [27]; habitat where this species is often encountered throughout its range [36, 37]. Re-sighting rates of individuals remain extremely low, which, combined with the seasonality of sightings, suggests a large transient population which

utilises habitat away from the reef systems of the Maldives. Where the Maldives *M. birostris* originate from, or travel to, remains unknown [35].

Manta rays are often observed in association with hitchhiker fish species, such as the golden trevally (*Gnathanodon speciosus*) and members of the remora family (Echeneidae) that closely follow (within 1 m) or attach themselves to their manta ray host [15, 23]. It has been suggested that hitchhiking behaviour evolved as a means to gain protection from predation, enhance foraging opportunities, increase locomotor efficiency, and increase encounters with conspecifics [15, 17, 38–42]. However, investigations into these hitchhiker associations are limited, and the links between interspecific interactions are sensitive to the abiotic environment in which they occur [5, 43, 44].

Here, we explore the spatial and temporal variation in the presence of hitchhikers with *M. alfredi* and *M. birostris* throughout the Maldives and investigate the factors which may influence association. This study aims to improve our ecological understanding of interactions between manta rays and their hitchhikers by highlighting how these associations are structured, and what the drivers of the associations might be [11].

## Materials and methods

### Study area

The Maldives archipelago is comprised of 26 geographical coral atolls and approximately 2,000 islands situated predominantly in the northern Indian Ocean (Fig 1) [45]. The research was carried out under permit from the Maldives' government (annually renewable permit: PA/2020/PSR-M07).

### Manta ray sightings and hitchhiker species

The unique ventral body pigmentation of each manta ray enables individuals to be distinguished from one-another using a photo-identification (photo-ID) catalogue of the ventral surface of the rays [26, 27, 46]. A manta ray sighting was defined as a confirmed photo-ID of an individual manta ray on a given day at a specific location. Surveys were performed via SCUBA or freediving by trained Manta Trust staff (www.mantatrust.org) and citizen science contributors between 1987–2019. Surveys were carried out across the whole archipelago throughout the year, of all study years, although known *M. alfredi* and *M. birostris* aggregation sites were surveyed most frequently, creating some sampling bias.

Where possible each manta ray was identified to species [47], and the sex and maturity status (adult, subadult, juvenile) of each was recorded during the dive/snorkel. For each sighting, the pregnancy status (an estimate of trimester) and primary behavioural activity (cleaning, feeding, courtship, cruising, or breaching) were also recorded [27, 48]. Thereafter, two-step verification and further sighting details were determined through assessment of the image/s by trained staff using the methods described in Stevens [27], and Peel *et al.* [49]. For this study, manta ray sightings were removed from the analysis if the sex or maturity status of the ray could not be determined.

The presence, number, and species of hitchhikers associated with each manta ray sighting were recorded during the dive/snorkel, and verified by visual analysis of all photo-ID images (Fig 2), using FishBase to determine species identification [50]. The conspicuous colour, patterns and behaviour of cleaner fish (Labridae) enabled them to be clearly distinguished from the hitchhiker species [51]. Identified sharksucker remora (*Echeneis naucrates*) were further classified as either adults (>20 cm) or juveniles (≤20 cm) based on visual estimates against the host size [39], as well as differences between their colour and

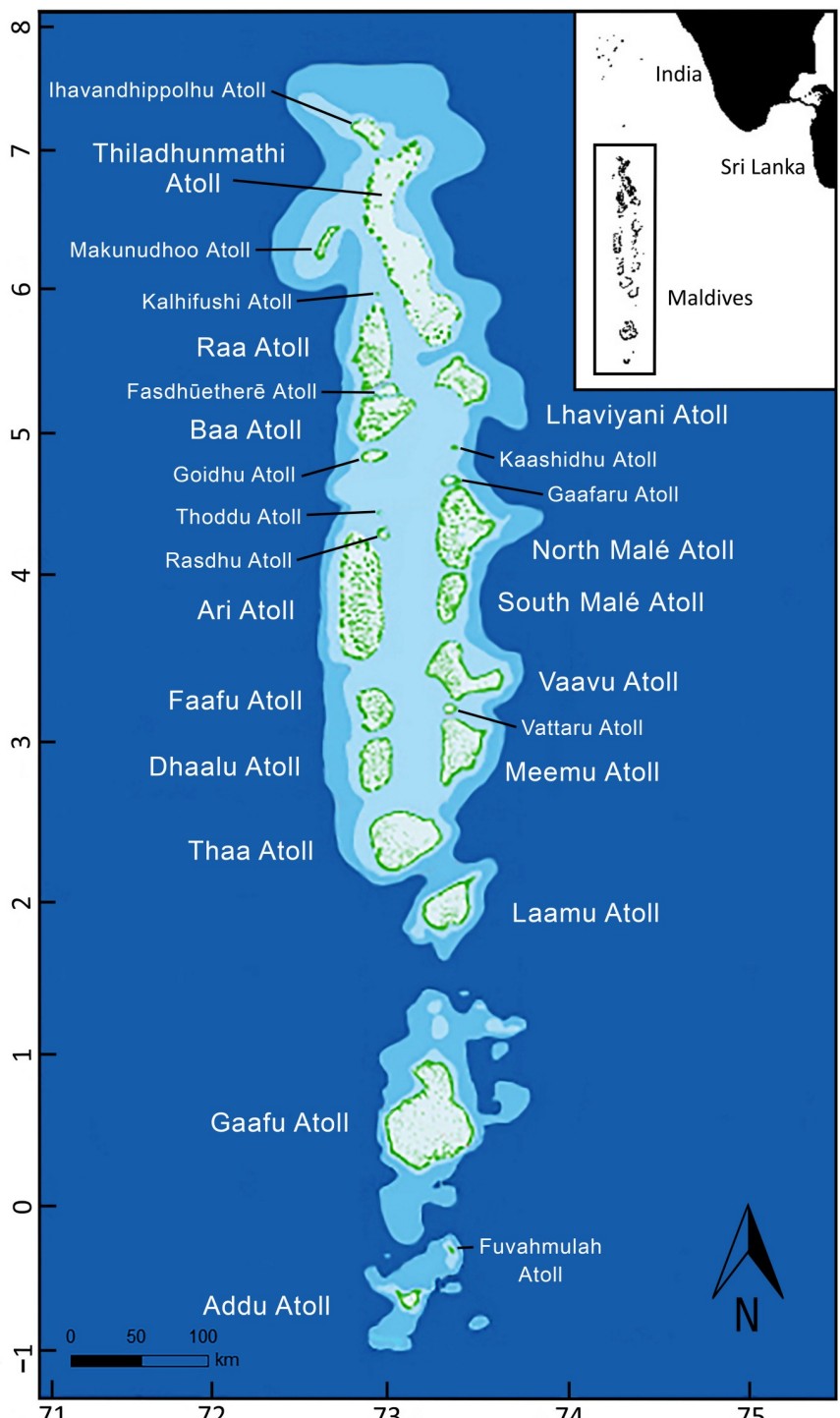

**Fig 1. A map of the Maldives archipelago located to the southwest of India.** Diagram shows the 26 geographical atolls illustrated in green.

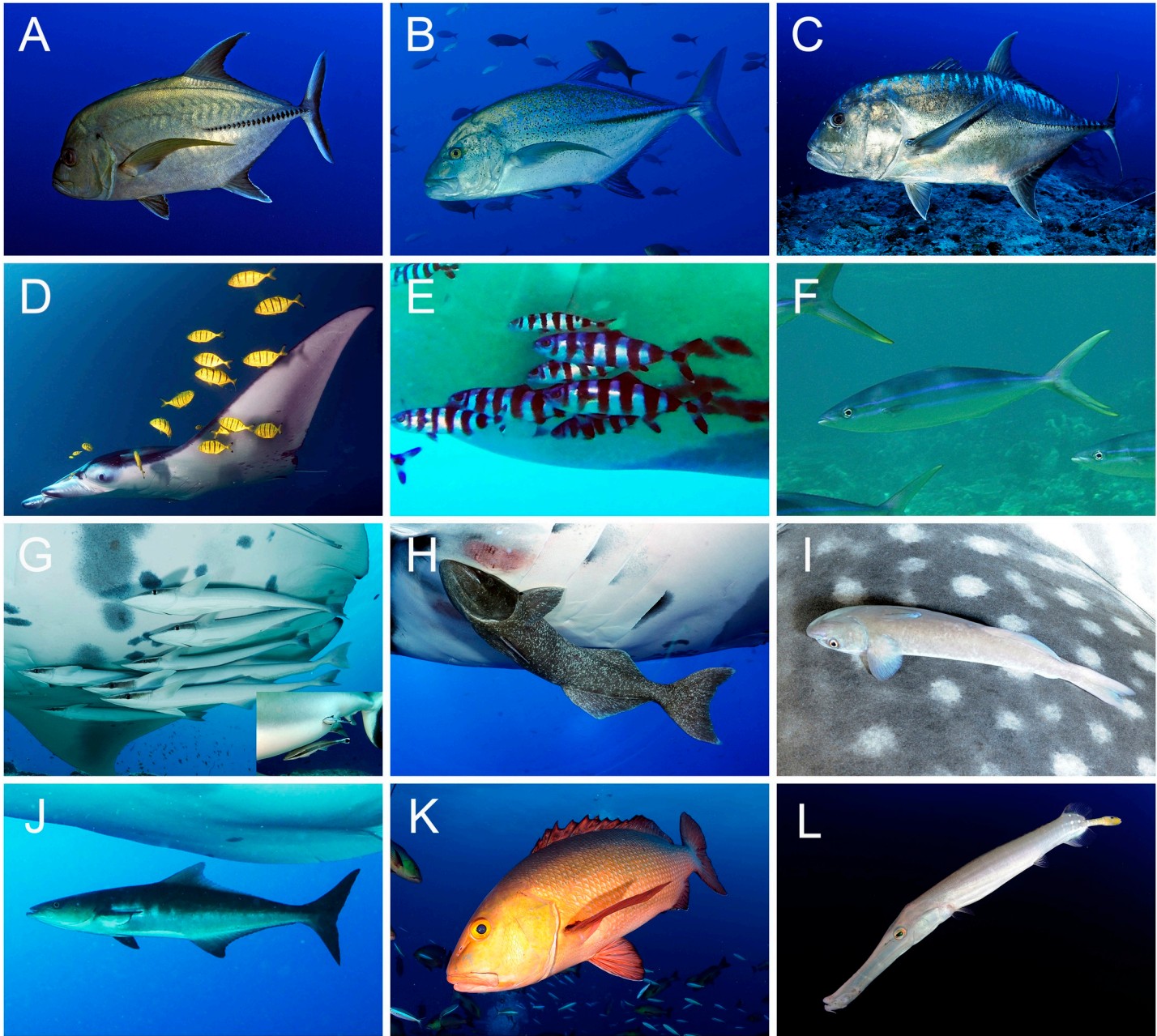

**Fig 2. Images of hitchhiker species used for identification.** (A) black trevally (*Caranx lugubris*), (B) bluefin trevally (*Caranx melampygus*), (C) giant trevally (*Caranx ignobilis*), (D) golden trevally (*Gnathanodon speciosus*), (E) pilot fish (*Naucrates doctor*), (F) rainbow runner (*Elagatis bipinnulata*), (G) sharksucker remora (*Echeneis naucrates*) (juvenile inset), (H) giant remora (*Remora remora*), (I) little remora (*Remora albescens*), (J) cobia (*Rachycentron canadum*), (K) red snapper (*Lutjanus bohar*), and (L) Chinese trumpetfish (*Aulostomus chinensis*). All images © The Manta Trust.

body patterns (Fig 2). Any hitchhiker species which could not be identified (due to poor image quality), were removed from the analysis. Each site utilised by *M. alfredi* was classified by site function (feeding area, cleaning station or cruising area) based on the predominant behaviour observed at the location [34]. Almost all *M. birostris* observed were cruising, therefore no site function was investigated.

## Data analysis

**Variations in hitchhiker observations.** Manta ray (*M. alfredi* and *M. birostris*) sightings and the total number of sightings where associated hitchhiker species were observed were summarised. To investigate variations in the total number of the most frequently observed hitchhiker species present with *M. alfredi*, the difference in the daily mean number of each hitchhiker species (number of sightings / number of hitchhiker species observed) was compared among manta ray sex and pregnancy stage (male, non-pregnant female, 2$^{nd}$ trimester pregnant female, 3$^{rd}$ trimester pregnant female, and 4$^{th}$ trimester pregnant female), season (NE or SW Monsoon), maturity status (adult, subadult, and juvenile), and site function (cleaning station, feeding area, cruising area). For *M. birostris*, categories included sex (male or female), season, and maturity status. Due to unequal sample sizes, and violation of the homogeneity of variance assumption, a Welch's ANOVA was used followed by a Games-Howell post-hoc test using the 'oneway.test' function of the 'Rmisc' R package, and 'oneway' function of the 'userfriendlyscience' R package, respectively [52].

**Spatial and temporal variation in hitchhiker presence.** Spatial variation in the presence of the most frequently observed hitchhiker species with *M. alfredi* (adult and juvenile *E. naucrates*) were investigated by mapping the percentage of sightings at each site (grouped by site function) where the hitchhiker species was present (total number of sightings where hitchhikers were observed / total number of sightings at the site) in ArcGIS 10.7. Any sites with a total of nine or fewer sightings (213 sites) were excluded to reduce the bias a low number of sightings may have on analysis.

Temporal variation in the presence of adult and juvenile *E. naucrates* with *M. alfredi* was investigated using monthly time series. This series incorporated the period with the greatest number of sightings (2008–2019) to provide a suitable period from which to visualise trends (i.e., seasonality). The monthly total number of sightings were corrected for survey effort by calculating the mean monthly number of manta rays observed per survey (monthly total manta ray sightings / monthly total number of surveys).

**Generalised linear mixed models.** Logistic generalised linear mixed models (GLMM) using R v4.0.0 [52] were used to investigate relationships between the presence of the most frequently observed hitchhiker species (adult and juvenile *E. naucrates*, *G. speciosus, and Lutjanus bohar*) with *M. alfredi* and four explanatory variables: sex with pregnancy status, maturity status, site function (determined by the predominant behaviour observed at the site [34]), and seasonality (NE or SW Monsoon). Due to the low number of recorded associations between *M. alfredi* and most of the hitchhiker species, only those with sufficient data were included in the GLMM analysis. The same model was used for *Remora remora* (the most frequently observed hitchhiker species with *M. birostris*), but without site function, and sex was classified only as male or female as pregnancies were only observed during four sightings. Each GLMM was fitted with a logit link function to the binary response of hitchhiker species presence (1) and absence (0) using the 'lme4' R package [53]. Each model contained the manta-ID as a random intercept to account for any temporal autocorrelation arising from individual rays being repeatedly observed [54]. To compare the relative goodness-of-fit, GLMM models without random effects (GLM) were tested. To reliably estimate the parameters, categories of variables with levels observed equal to or less than five times were removed. For example, under the category behavioural activity, the level 'breaching' was observed on less than five occasions, so was removed from analysis. The most informative explanatory variables were identified by firstly testing GLMM models with all combinations of explanatory variables. The variance inflation factor (VIF) was used to test models for multicollinearity; the maximum VIF was <1.5. Model performance was assessed using corrected Akaike information criterion (AICc)

test statistic [55] using the 'MuMin' R package [56], and the DHARMa R package [57] was used to check the model residuals were normally distributed. The highest-ranking models (with the lowest AICc value, S1 Table) for each hitchhiker species were then interpreted in terms of odds ratios (ORs) (the likelihood of the presence of the hitchhiker species in comparison with the reference category). Any models with ΔAICc <2 were considered in interpretation of the highest-ranking model [55]. The significance of each explanatory variable was determined by the 95% confidence interval (CI) of OR, whereby a narrower CI indicates a more precise estimation while, in comparison, a wider CI which had a greater uncertainty. A CI that crossed one is considered non-significant. Any ORs with $p > 0.05$ are not reported.

## Results

### Reef manta ray (*M. alfredi*) sightings and associated hitchhiker species (1987–2019)

A total of 4901 *M. alfredi* were individually identified [male = 2442 (50%), female = 2459 (50%)] during a total of 72912 sightings, of which 44071 (60%) were of females [adult = 25700 (58%), juvenile = 18371 (42%)] and 28841 (40%) were males [adult = 25968 (90%), sub-adult = 1443 (5%), juvenile = 1430 (5%)]. All sightings occurred across 353 sites, of which 95 (27%) were cleaning stations [sightings = 24034 (33%)], 53 (15%) were cruising areas [sightings = 129 (0%)], and 205 (58%) feeding areas [sightings = 48749 (67%)].

Twelve different species of hitchhiker were identified with *M. alfredi* (Table 1 and Fig 3). The most frequently observed hitchhiker species with *M. alfredi* was *E. naucrates*. Adult *E.*

**Table 1. Summary of hitchhiker species observed with manta rays.**

| Manta ray species | Hitchhiker species | Total no. hitchhiker individuals observed | No. ray sightings when observed | Max no. observed per sighting | Mean no. per sighting ±SD |
|---|---|---|---|---|---|
| *Mobula alfredi* | *Caranx melampygus* | 53 | 44 | 9 | 1.2 ± 1.2 |
| | *Caranx ignobilis* | 26 | 26 | 1 | 1 ± 0 |
| | *Gnathanodon speciosus* | 1176 | 536 | 29 | 2.1 ± 3.2 |
| | *Naucrates doctor* | 9 | 6 | 3 | 1.5 ± 0.8 |
| | *Elagatis bipinnulata* | 28 | 25 | 3 | 1.2 ± 0.4 |
| | *Echeneis naucrates* | 16549 | 8211 | 24 | 2 ± 1.4 |
| | *Echeneis naucrates* (juv.) | 1025 | 967 | 4 | 1.1 ± 0.3 |
| | *Remora remora* | 1 | 1 | 1 | 1 |
| | *Remora albescens* | 41 | 40 | 2 | 1 ± 0.2 |
| | *Rachycentron canadum* | 18 | 18 | 1 | 1 ± 0 |
| | *Lutjanus bohar* | 247 | 228 | 3 | 1.1 ± 0.3 |
| | *Aulostomus chinensis* | 3 | 3 | 1 | 1 ± 0 |
| *Mobula birostris* | *Caranx lugubris* | 31 | 25 | 3 | 1.2 ± 0.5 |
| | *Naucrates doctor* | 9 | 1 | 9 | 9 |
| | *Echeneis naucrates* | 6 | 2 | 3 | 3 ± 0 |
| | *Echeneis naucrates* (juv.) | 2 | 2 | 1 | 1 ± 0 |
| | *Remora remora* | 582 | 398 | 3 | 1.5 ± 0.5 |

Total and mean number of hitchhikers observed with *Mobula alfredi* between 1987–2019 (*total sightings* = 72912) and *Mobula birostris* between 1996–2019 (*total sightings = 726*).

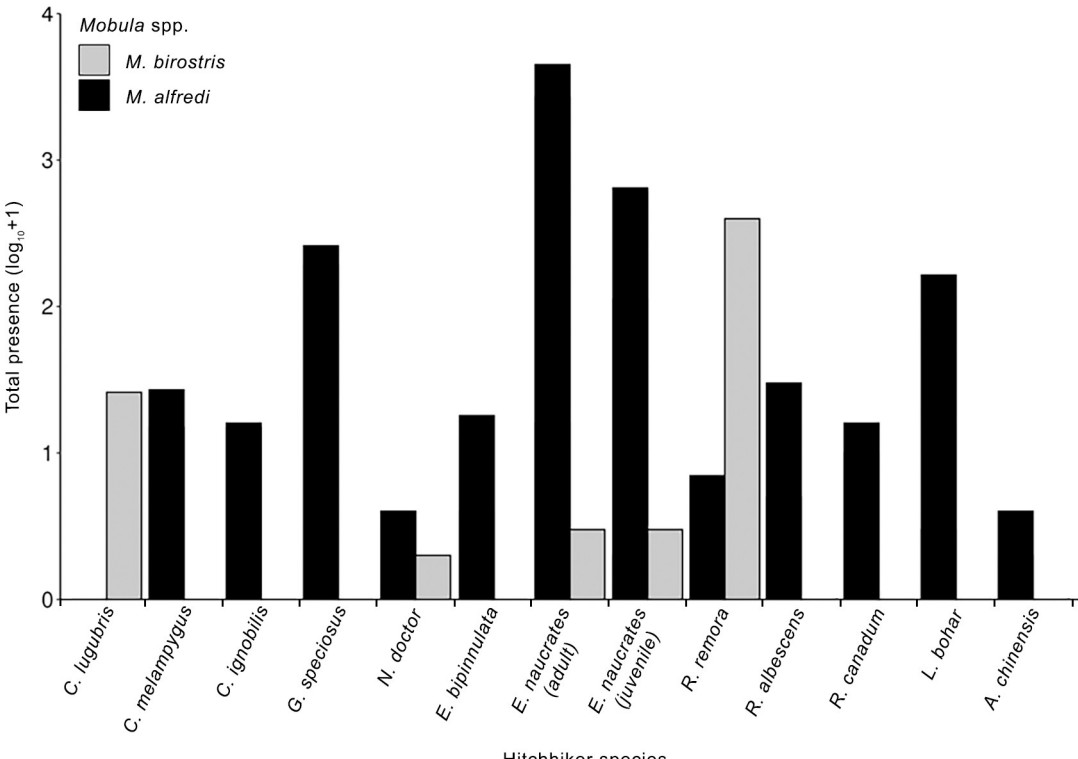

**Fig 3. Total presence of hitchhiker species observed with manta rays.** The total number of sightings where each identified hitchhiker species ($n$ = 12) was observed with *Mobula alfredi* (black) or *M. birostris* (grey). A (10+1) transformation was used for better visualisation of the data.

*naucrates* were observed with *M. alfredi* during 7189 (10%) of the total sightings, and juveniles during 756 (1%) sightings.

## Variations in hitchhiker associations with reef manta rays (*M. alfredi*)

Due to the low number of recorded associations between *M. alfredi* and most of the hitchhiker species, only *E. naucrates* was investigated further here.

**Adult sharksucker remora (*E. naucrates*).** When present, the number of adult *E. naucrates* associated with *M. alfredi* ranged between one and twenty-four individuals per sighting. There was a significant difference in the daily mean number of adult *E. naucrates* associated with *M. alfredi* between manta ray sex and pregnancy stage ($F_{4, 1302}$ = 102.6, $p < 0.001$). The highest mean number of *E. naucrates* occurred with female *M. alfredi* in their 4th trimester of pregnancy; significantly higher ($p < 0.001$) than with males, non-pregnant females, and 2nd trimester pregnant females (Fig 4). There was also a significant difference between maturity status categories ($F_{2, 3280}$ = 123.1, $p < 0.001$), where the highest mean number of *E. naucrates* were observed with adults, which was significantly higher ($p < 0.001$) than with subadults and juveniles (Fig 4). The total mean number of adult *E. naucrates* with *M. alfredi* was also significantly different between site functions ($F_{2, 1223}$ = 222.1, $p < 0.001$), which was significantly higher ($p < 0.001$) at cleaning stations and cruising areas than at feeding areas (Fig 4). The total mean number of adult *E. naucrates* associated with *M. alfredi* was also significantly different during each season ($F_{1, 6431}$ = 155.7, $p < 0.001$), with more *E. naucrates* associated with *M. alfredi* during the NE Monsoon ($p < 0.001$) (Fig 4).

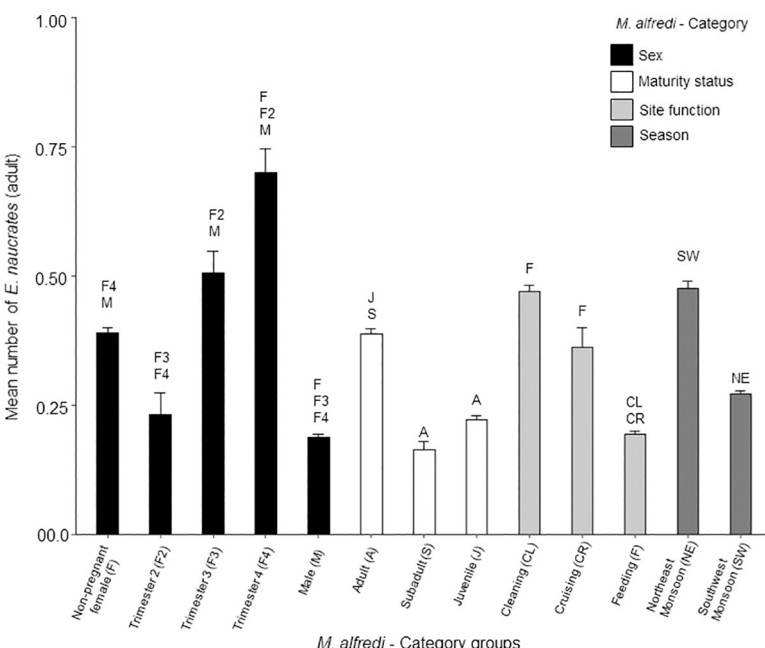

**Fig 4. Daily mean number of adult *Echeneis naucrates* (+SE) observed with *Mobula alfredi* between category groups.** Each category is coloured as per legend with group name below each bar. Letters above each bar correspond to those in brackets after the group name and indicate the groups with a significant difference (*p* < 0.001).

**Juvenile sharksucker remora (*E. naucrates*).**   When present, the number of juvenile *E. naucrates* associated with *M. alfredi* ranged between one and four individuals per sighting. There was a significant difference in the daily mean number of juvenile *E. naucrates* observed between manta ray sex and pregnancy stage ($F_{4, 1496}$ = 19.7, *p* < 0.001). The highest daily mean number of juveniles occurred with male *M. alfredi*, which was significantly higher (*p* < 0.001) than pregnant females (Fig 5). There was a significant difference in the maturity status category ($F_{2, 2661}$ = 15.6, *p* < 0.001), where the highest mean number of juvenile *E. naucrates* were observed with juvenile *M. alfredi;* significantly higher than with adults (*p* < 0.001). The mean number of juvenile *E. naucrates* associated with *M. alfredi* was also significantly different between site functions ($F_{2, 1204}$ = 30.3, *p* < 0.001), with a significantly higher (*p* < 0.001) amount at feeding areas than at cleaning stations. The daily mean number of juvenile *E. naucrates* associated with *M. alfredi* was also significantly different during each season ($F_{1, 5828}$ = 19.3, *p* < 0.001), with a significantly higher number of *E. naucrates* associated with *M. alfredi* during the NE Monsoon (*p* < 0.001).

**Spatial and temporal variation in hitchhiker presence.**   There was a total of 72361 sightings across 149 sites which had ten or more manta ray sightings; 23926 (33%) occurred at cleaning stations where *E. naucrates* were present during 4474 (19%) of observations, and 48435 (67%) occurred at feeding areas where *E. naucrates* were present during 2641 (5%) of observations. The highest percentage of *E. naucrates* presence by atoll occurred within Meemu (40%), North Malé (27%), and Ari (27%) Atolls. The sites with the highest percentage of *E. naucrates* present were Maayafushi Falhu (60%), a feeding area in Ari Atoll, at Rangali Madivaru (50%), a cleaning station in Ari Atoll, and at Delidhoo (50%), a cleaning station in Thiladhunmathi Atoll. *Echeneis naucrates* were not observed with *M. alfredi* at 25 sites, 17 of which are feeding areas (total sightings = 1512) and 8 were cleaning stations (total sightings = 266).

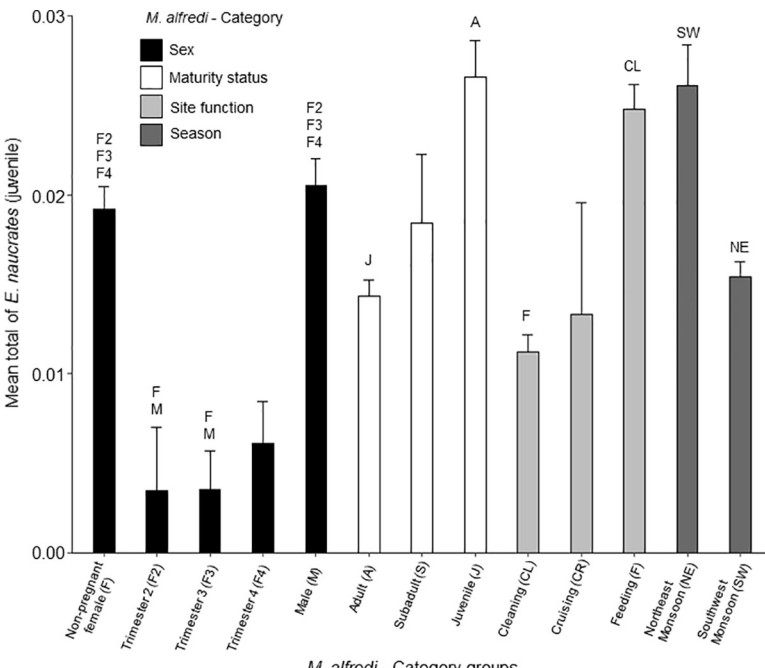

**Fig 5. Daily mean number of juvenile *Echeneis naucrates* (+SE) observed with *Mobula alfredi* between category groups.** Each category is coloured as per legend with group name below each bar. Letters above each bar correspond to those in brackets after the group name and indicate the groups with a significant difference ($p < 0.001$).

The mean percentage of *E. naucrates* present by site function was 11.6±12.2% at feeding areas, and 12.8±14.4% at cleaning stations (Fig 6).

The proportion of adult *E. naucrates* observed with *M. alfredi* was highest during the NE Monsoon, typically from January to March, at which time the lowest monthly mean number of *M. alfredi* sightings occurred (Fig 7). There was no seasonal pattern observed in the presence of juvenile *E. naucrates* with *M. alfredi*.

## Reef manta ray (*M. alfredi*) generalised linear mixed models

**Adult sharksucker remora (*E. naucrates*).** The highest ranking GLMM for *E. naucrates* contained all four predictors (site function, sex, maturity, and season) (S1 Table). The model (Fig 8) suggests that *E. naucrates* were most likely to be present with *M. alfredi* at cleaning stations, which were 49% more likely than at feeding areas (OR = 0.51). This hitchhiker species was also most likely to be present on females in their 4th trimester of pregnancy (OR = 2.6), which was 160% higher than non-pregnant females (reference category), while males (OR = 0.59) were 41% less likely to have *E. naucrates* hitchhikers than non-pregnant females. *Echeneis naucrates* were least likely to be present on juvenile *M. alfredi* (OR = 0.67), which were 31% less likely to have this hitchhiker species than adults (reference category). The GLMM also indicates that the likelihood of *E. naucrates* presence with *M. alfredi* was highest during the NE Monsoon (reference category), which was 25% more likely than during the SW Monsoon (OR = 0.75).

**Juvenile sharksucker remora (*E. naucrates*).** The highest ranking GLMM for juvenile *E. naucrates* contained all four predictors (site function, sex, maturity, and season). The model (Fig 8) suggests that juvenile *E. naucrates* were most likely to be present with *M. alfredi* at feeding areas (OR = 2.08), which was 108% more likely than at cleaning stations (reference

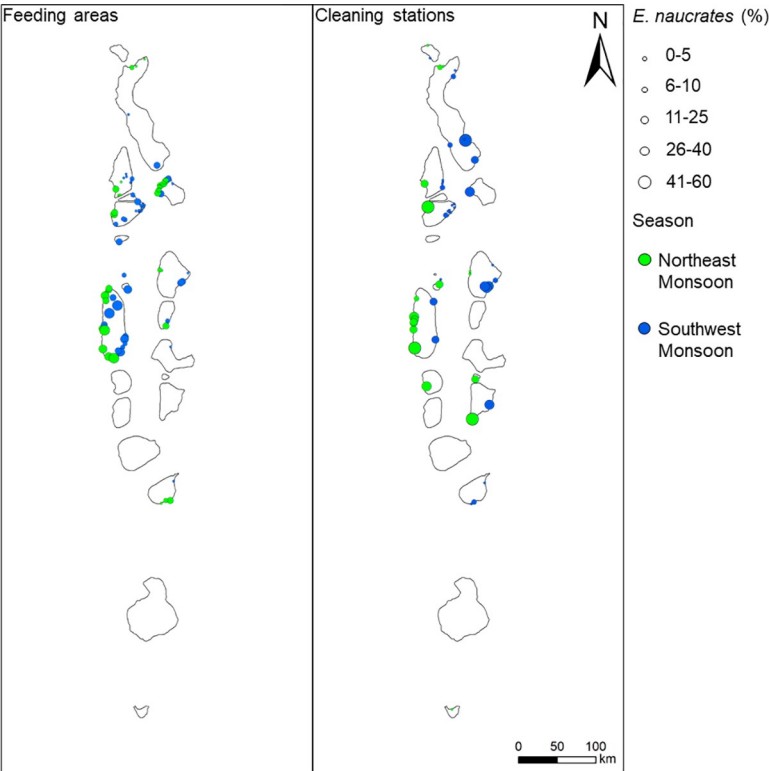

**Fig 6. Heatmaps coloured by season and percentage of sightings where *Echeneis naucrates* were present.** Includes feeding areas and cleaning stations with > 10 *E. naucrates* sightings.

category). Juvenile *E. naucrates* were also more likely to be present on juvenile *M. alfredi* (OR = 2.37), which were 137% more likely to have this hitchhiker group than adults (reference category). Male *M. alfredi* (OR = 1.59) were 59% more likely than females to have juvenile *E. naucrates* present, and juvenile *E. naucrates* were least likely to be present with *M. alfredi* during the SW Monsoon (OR = 0.63); 37% less likely than during the NE Monsoon (reference category).

**Red snapper (*L. bohar*).** The highest ranking GLMM for *L. bohar* (Fig 8) contained one significant predictor (site function) and suggests that the species were most likely to be present with *M. alfredi* at cleaning stations (reference category), which was 100% more likely than at feeding areas (OR = 0). There was one model with ΔAICc <2 (S1 Table), which was the same

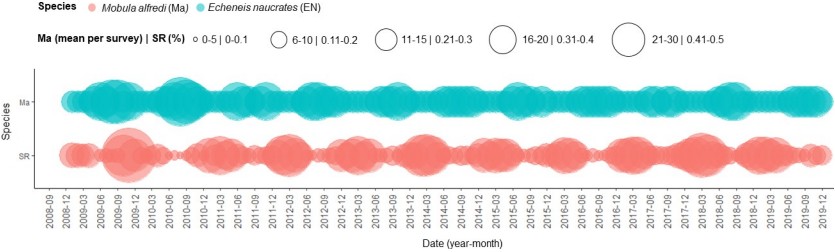

**Fig 7. Time series plot showing *Mobula alfredi* sightings and *Echeneis naucrates* presence.** The total monthly number of *Mobula alfredi* sightings 2008–2019, and the percentage of those that had *Echeneis naucrates* associations.

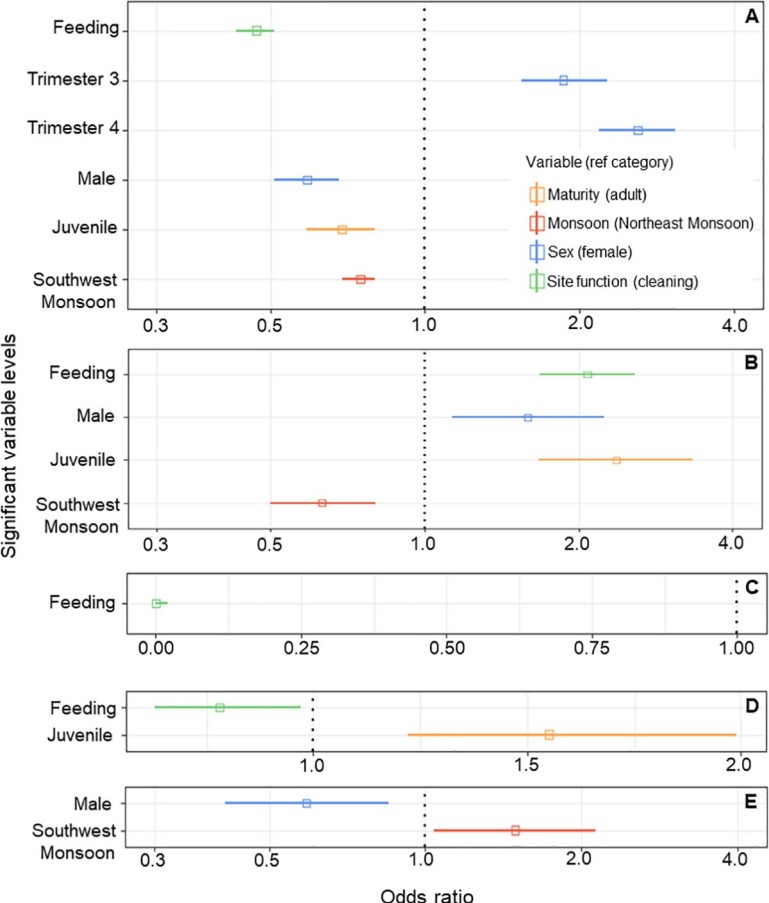

**Fig 8. Relationship between hitchhiker species presence and significant explanatory variables ($p < 0.05$) in terms of odds ratio (OR).** Indicates the likelihood of presence in comparison with the reference category shown in the legend. OR values are plotted with 95% confidence intervals (CI; solid horizontal lines). Where the CI does not span 1, the explanatory variable is significantly more likely when OR > 1, and significantly less likely when OR < 1. (A) *Mobula alfredi* and *Echeneis naucrates*, (B) *M. alfredi* and juvenile *E. naucrates*, (C) *M. alfredi* and *Lutjanus bohar*, (D) *M. alfredi* and *Gnathanodon speciosus* presence, and (E) *M. birostris* and *Remora remora*.

as the highest-ranking model with the addition of season, but this predictor was non-significant ($p > 0.05$).

**Golden trevally (*G. speciosus*).** The highest ranking GLMM for *G. speciosus* contained two significant predictors (site function and maturity status). *Gnathanodon speciosus* were most likely to be present with *M. alfredi* at cleaning stations (reference category), which was 22% more likely than at feeding areas (OR = 0.78) (Fig 8). There were two models with ΔAICc <2 (S1 Table). One of these models contained the same predictors as the highest-ranking model with the addition of season, but this predictor was non-significant ($p > 0.05$). The other model contained only the predictor maturity status, which suggested *G. speciosus* were more likely to be present on juvenile *M. alfredi* (OR = 1.5), which were 55% more likely to have this hitchhiker species than adults (reference category).

## Oceanic manta ray (*M. birostris*) sightings and associated hitchhiker species

A total of 663 *M. birostris* were individually identified [male = 363 (55%), female = 300 (45%)] during a total of 726 sightings, of which 329 were females [adult = 237 (72%), subadult = 24

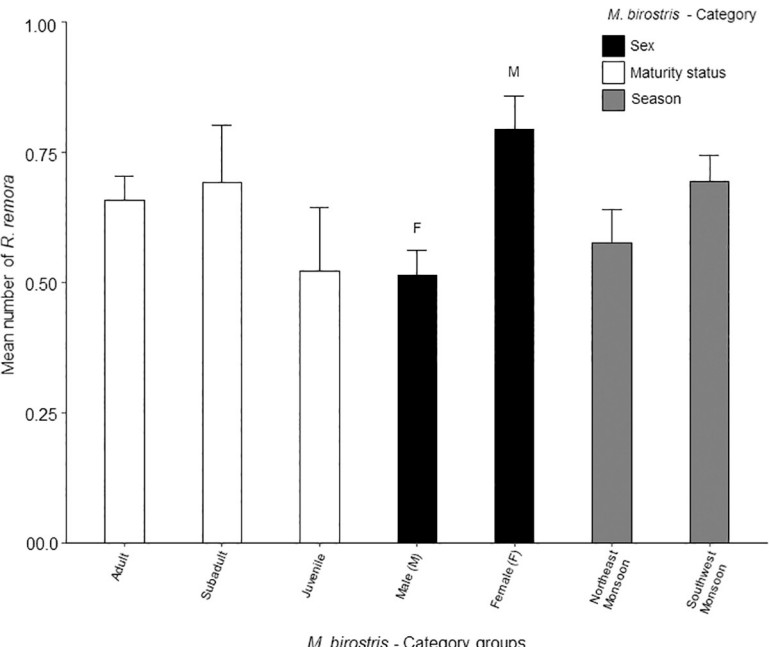

**Fig 9. Daily mean number of *Remora remora* (+SE) observed with *Mobula birostris* between category groups.** Each category is coloured as per legend with group name below each bar. Letters above each bar correspond to those in brackets after the group name and indicate the groups with a significant difference ($p < 0.001$).

(7%), juvenile = 68 (21%)] and 397 were males [adult = 371 (93%), subadult = 24 (6%), juvenile = 2 (1%)]. All sightings occurred across 39 sites, of which, 642 (88%) occurred at Fuvahmulah Atoll, and 662 (91%) occurred during the months of March and April, straddling the transition between the NE and SW Monsoons. *Mobula birostris* were observed exhibiting cleaning [sightings = 8 (1%)], cruising [sightings = 681 (94%)], feeding [sightings = 10 (1%)], and courtship behaviour [sightings = 27 (4%)].

Five different hitchhiker species were identified associated with *M. birostris* (Table 1 and Fig 3). The most frequently observed hitchhiker species was *R. remora*, which was observed with *M. birostris* during 397 (55%) sightings.

**Variations in hitchhiker associations with oceanic manta rays (*M. birostris*).** Due to the low number of recorded associations between *M. birostris* and most of the hitchhiker species, only *R. remora* was investigated further here.

When present, the number of *R. remora* associated with *M. birostris* ranged between one and three individuals per sighting. The highest daily mean number of *R. remora* occurred with female *M. birostris*, which was significantly higher than males ($F_{1, 233} = 12.7$, $p < 0.001$). There were no significant differences for *M. birostris* maturity status, or between seasons (Fig 9).

**Oceanic manta ray (*M. birostris*) generalised linear mixed models.** *Remora remora* were most likely to be present with female *M. birostris* (reference category); 41% (OR = 0.59) more likely than with males (Fig 8), and they were more likely to be present during the SW Monsoon which was 49% higher (OR = 1.49) than during the NE Monsoon (reference category) (Fig 8). There were two models with ΔAICc <2 (S1 Table). Both models contained the same predictors as the highest-ranking model (sex and season) with the addition of maturity status. This GLMM model suggests *R. remora* may also be most likely to be present with adult *M. birostris* (reference category); 47% (OR = 0.53) more likely than with juveniles.

## Discussion

A variation in the species of hitchhiker associated with *M. alfredi* and *M. birostris* was identified, with *E. naucrates* and *R. remora* being the most common, respectively. Spatiotemporal variation in the presence of manta rays was identified as a driver for the occurrence of ephemeral hitchhiker associations, and for the first time, associations between *M. alfredi* and *M. birostris* with hitchhiker species other than those belonging to the family Echeneidae are described.

### Reef manta ray (*M. alfredi*)

Twelve hitchhiker species were observed associating with *M. alfredi*, of these, *E. naucrates* was by far the most frequent. The Echeneidae family are well-known for their hitchhiking behaviour on large-bodied vertebrates [23, 39, 41]. The relationship with their hosts is generally considered mutualistically symbiotic, as most remora species spend all their post-larval life in close association with their hosts, eating their ectoparasites in return for a range of benefits [15, 39, 40, 58]. However, the degree of host specificity and the nature of the association has been shown to vary along the symbiotic continuum (mutualism, commensalism, and parasitism), and the importance of this host food source varies between remora species and at different life stages [1, 25, 38, 39]. Recent investigations into the echeneid-host association also suggest that there may be significant costs incurred by the host, such as persistent damage and scarring from the adhesive disc by which the remoras attach themselves to their hosts, as well as deforming injuries when smaller-sized remoras force themselves through the gill slits and other body openings of their mobulid host [15, 23, 59].

*Echeneis naucrates* is a neritic species that is presumed to be physiologically unable to remain attached to their host when they dive at depth [15, 22, 58]. *Mobula alfredi* in the Maldives and elsewhere throughout their range are known to undertake regular dives below 200 m, presumably to forage on prey within the deep-scattering layer [33, 60–63]. Consequently, we suggest *E. naucrates* associations with their *M. alfredi* hosts are often ephemeral, with adults spending significant periods of time free-swimming, re-associating with a host upon their return to the remora's habitat [15, 58]. Given other hosts of adult *E. naucrates* are also known to make regular dives below the Neritic Zone [64, 65], it is likely associations with these hosts are also ephemeral. Indeed, it is unknown what percentage of their adult lives *E. naucrates* spend away from their hosts, but it is not uncommon for this species to be observed free-swimming, or resting on the seabed in groups, during reef dives in the Maldives (Stevens, *pers. obs.*).

*Echeneis naucrates* associations with *M. alfredi* varied significantly depending on host sex, pregnancy state, and maturity status. Associations also varied significantly depending on the function of the observation site and the season within which the sighting occurred. *Echeneis naucrates* were significantly more likely to be associated with female *M. alfredi* than males, and with *M. alfredi* at cleaning stations. Previous studies have demonstrated that female *M. alfredi* are significantly more likely to be sighted at cleaning stations than males [27, 66]. Cleaning stations are predominantly located on shallow reefs or in lagoons [27]; suitable habitat for the neritic *E. naucrates* [58]. Therefore, more frequent utilisation of cleaning stations by female *M. alfredi* provides greater opportunity for an association to occur, and the more time spent at these sites, the greater the opportunity for a higher total number of associations to occur. In contrast, many shallow *M. alfredi* feeding sites in the Maldives, and elsewhere, are situated in atoll channels which are adjacent to deep-water areas, where it is hypothesised *M. alfredi* forage prior to shallow-water feeding events [67]. As *E. naucrates* may be physiologically unable to remain associated with *M. alfredi* when they travel to these deep-water locations, the likelihood of association during subsequent shallow-water feeding is reduced [15, 67].

Near-term pregnant female *M. alfredi* had the highest likelihood of an association, and the highest mean number of *E. naucrates* associates per individual; consistent with a previous preliminary study in the region [68]. The study suggested these increased associations were a result of thermoregulatory advantages gained by pregnant *M. alfredi* occupying warm-water habitat for longer periods during late-term gestation to reduce gestation times [15, 68], a common behaviour documented in other elasmobranchs [69–71]. If this hypothesis is correct, it would explain, in part, why female *M. alfredi* generally are recorded more frequently at cleaning stations than males. And as stated above, the longer the continuous period an individual *M. alfredi* spends in suitable *E. naucrates* habitat, the greater the chance of an association/s occurring.

A seasonal reproductive strategy, something which has been observed in both captive [72] and wild-caught *E. naucrates* [22], is typically adopted by species whose access to resources, such as prey and host availability, is uncertain; aiming to maximise juvenile recruitment [22, 73]. Bachman *et al.* [22] highlighted that hitchhiker behaviour in terrestrial arthropods predicts that host association restricts mate selection, reproduction, and location. Therefore, echeneid population dynamics are likely to be strongly influenced by the ecology and availability of their hosts. For juvenile *E. naucrates*, host parasitic copepods comprise a more integral part of their diet than that of an adult [38], and juvenile remoras that are not attached to a host may be exposed to unsuitable environments and increased predation risk [58]. During the current study, juvenile *E. naucrates* were more likely to be associated with juvenile *M. alfredi*, most likely because juvenile manta rays spend most of their time in protected lagoons and other shallow water nursery habitats [27, 66], increasing the chance of long-term associations between the two species for reasons already discussed in this study. Thus, juvenile manta rays are likely to provide a more suitable host for the juvenile remoras that require continual host associations in shallow water to survive to adulthood [22, 27, 58]. The obligate symbiosis of juvenile *E. naucrates* with their hosts may then transition to a more facultative relationship in adult remoras [22, 58].

Associations between *E. naucrates* and *M. alfredi* were also significantly higher during the NE Monsoon; the first record of a seasonal trend in *E. naucrates* host association. The higher association rate during the NE Monsoon may potentially be associated with the suppression of primary productivity, which peaks during the Maldives' SW Monsoon [33, 34]. As proposed sites of behavioural thermoregulation and predator avoidance [15, 27], cleaning stations may be utilised more during periods of lower primary productivity to conserve energy and reduce risk of predation. Thus, the greater period a manta ray spends within such habitat, the greater chance of an association occurring with a *E. naucrates*. However, greater abundances of large skipjack (*Katsuwonus pelamis*) and yellowfin tuna (*Thunnus albacares*) have been observed in the Maldives in the less productive NE Monsoon [33]. This led Anderson *et al.* [33] to suggest that there may be an increase in primary productivity at this time associated with a deep chlorophyll maximum (DCM) that is not visible to the satellite technology (SeaWiFS) which records the high SW Monsoon productivity. Manta rays are poikilothermic, with an optimal thermal temperature of 20–26°C, but can endure colder temperatures for short periods due to a counter-current heat-exchange mechanisms [63, 74]. This physiological adaption enables manta rays to forage on zooplankton blooms within the deep-scattering layer, and down to depths of over 672 metres and temperatures of 7.6°C [63]. If Anderson *et al.* [33] hypothesis is correct, basking in warmer shallow waters during the NE Monsoon, at sites like cleaning stations, prior to and post deep forays may enable manta rays to physiologically prepare for, and recover from, the large metabolic costs incurred from such deep foraging bouts [63, 75]. This behavioural thermoregulation has also been used to explain why the spinetail devil ray (*Mobula mobular*) [76], the whale shark (*Rhincodon typus*) [75], and tuna species return to

shallow habitats after deep dives [77]. To support this hypothesis, more data on the diving behaviour and habitat use of manta ray in the Maldives is required, utilising camera-mounted and satellite tracking technologies, along with chlorophyll-α measurements, to address this question.

Considering the ecology of both the host and the symbiont, the results of this study suggest that the patterns of association between *E. naucrates* and *M. alfredi* are most likely driven by the spatiotemporal variation in presence of manta rays in the sharksucker's habitat. The overlap in species habitat-use at cleaning sites and within sheltered lagoons, particularly during the NE Monsoon, provides an explanation for why near-term pregnant female *M. alfredi* have the greatest likelihood of associating with adult *E. naucrates*, and why juvenile *M. alfredi* have the greatest likelihood of associating with juvenile *E. naucrates*.

Here, for the first time, associations between *M. alfredi* and hitchhiker species, other than those belonging to the family Echeneidae, are described. In reef fish systems, follower-feeding associations are influential on the structure of surrounding reef communities [11]. Competitive life in reef habitats brings about short-term associations to obtain food [11–13]. However, many ephemeral interactions between small teleost's and megafauna species, such as the one described here in the introduction between *R. canadum* and *R. bonasus* [16], are yet to be investigated. Considering the carnivorous feeding ecology of many of the non-echeneid hitchhikers identified here, such as the black (*Caranx lugubris*), bluefin (*C. melampygus*) [78], and giant (*C. ignobilis*) trevallies [79], as well as the rainbow runner (*Elagatis bipinnulata*) [80], cobia (*Rachycentron canadum*) [16], red snapper (*Lutjanus bohar*) [81], and Chinese trumpet-fish (*Aulostomus chinensis*) [82], it is likely these associates also opportunistically utilise the body of the manta ray to get near their prey [11–13, 82]. An observation supported by the authors in the field during this study for all the aforementioned species.

Several of the other hitchhiker associations with *M. alfredi*, such as the golden trevally (*Gnathanodon speciosus*) and the pilot fish (*Naucrates doctor*), are likely to be driven primarily by the advantage of the shelter provided by the host [14, 15]. In the case of *G. speciosus*, these associations only last until the juveniles are large enough to survive by themselves [15, 83]. Aside from *N. doctor* (which was only sighted with *M. alfredi* on six occasions during the study), all the non-remora hitchhiker species identified are neritic reef-dwellers [17, 84]. Therefore, there are likely to be more opportunities for associations to form between these species and *M. alfredi* than with the predominantly oceanic *M. birostris* [37]. This hypothesis is supported by the results of the GLMM models, which predicted the greatest chance of association rates between *M. alfredi*, *L. bohar* and *G. speciosus* at cleaning stations. Furthermore, as with juvenile *E. naucrates*, juvenile *G. speciosus* were significantly more likely to be associated with juvenile *M. alfredi*. The models also suggest that maturity status might be more influential than site function as one model with ΔAICc <2 contained only the predictor maturity status. It is possible the juvenile manta rays are similarly providing a more suitable host for the juvenile trevallies, which may also require continual host associations in shallow water to survive to adulthood [15, 27]. All of the non-echeneid hitchhiker species identified are considered to engage in commensalism, as they obtain benefits whilst their mobulid host receives neither benefits or harm from the association [1, 15–18, 83]. Overall, the hitchhiker species which associated most with *M. alfredi* reflect the characteristic assemblage of species that occupy the habitat where the host spends time [21].

The lower number of associations of non-echeneid hitchhiker species with *M. alfredi* could be a result of the analysis techniques used in this study, as sightings images were primarily focused on the ventral surface of the ray (to get a suitable photo-ID), but species such as *G. speciosus* often reside around the head of their host, exhibiting 'piloting' behaviour [15, 83]. Similar sampling bias is also likely to explain the low number of sightings where little remora

(*Remora albescens*) were present (*n* = 40) in this study [15]. This cryptic and poorly studied remora species was rarely observed outside the body of the manta rays. However, opportunistic visual inspection inside the buccal cavity of feeding adult *M. alfredi* by the authors during the study often revealed a pair of this species attached to the upper cavity. Therefore, most associations between these two species during this study were probably missed but were likely common.

## Oceanic manta ray (*M. birostris*)

Five hitchhiker species were found to be associated with *M. birostris*. Of these, *R. remora* was by far the most frequently observed, unlike in *M. alfredi*, where only one association was recorded between these two species during this study. And unlike the *M. alfredi* and adult *E. naucrates* association, the symbiosis between *M. birostris* and *R. remora* appears long-term, with the remora rarely, if at all, leaving the protection of its host [15, 23, 38], even when feeding at 130–140 m depth [37]. Previous examination of the diet data revealed that parasitic copepods comprise a crucial part of the *R. remora* diet, but that the importance decreases as the remora increases in size [25, 38, 39]. Despite this, the obligate nature of *R. remora*, like that of juvenile *E. naucrates* [58], further suggests that echeneid population dynamics are influenced by the distribution patterns of its host [22, 58].

In Mexico's Revillagigedo Archipelago's National Park, where the only other peer-reviewed study of manta hitchhiker associations has been undertaken, Becerril-García *et al.* [23] found no association between the total number of *R. remora*, *M. birostris* sex, morphotype, and month of the year. As a result, it was suggested that the presence of remoras could be influenced by the level of host ectoparasites, population size, diving behaviour, and surrounding environmental conditions. In the present study, a mean number of 1.5 ± 0.5 *R. remora* were observed per *M. birostris* sighting. In Mexico, a mean number of 1.6 ± 0.6 *R. remora* attached to each manta were observed [23], in Peru there has been a sighting of eleven echeneids associated with a single *M. birostris* [85], and at Isla de la Plata in Ecuador, as many as 40 individual *R. remora* have been recorded associating with a singe *M. birostris* (Guerrero, pers. comm.; Harty pers. comm.). Indeed, at Isla de la Plata, large numbers of *R. remora* are frequently associated with their manta hosts. Regional ecological variations between *M. birostris* populations, as well as the surrounding environmental conditions, are likely to be influencing the presence of *R. remora*. The significant differences in *R. remora* association rates recorded between male and female *M. birostris* in this study may also be linked to either foraging or reproductive strategies [35], although much more knowledge of *M. birostris* habitat use and behavioural ecology is required to address this hypothesis satisfactorily. Therefore, research into the ecological variations within and between *M. birostris* populations is a topic worthy of future research and could reveal valuable insight into the ecology of both the host and the symbiont [23].

Due to logistical constraints, accurate size data on the sighted manta rays during this study was rarely collected. Therefore, variation in manta ray disc width (DW) and hitchhiker presence was not investigated here. However, future directed studies could investigate DW (i.e., the sexual dimorphism between male and female manta rays) as a potential further pattern of association. Furthermore, the potential hosts costs and benefits of different echeneid densities remains relatively unexplored [59]. Research into these factors could also provide further insight into the patterns of association identified within the current study.

Unlike *E. naucrates*, *R. remora* frequently attach themselves to the dorsal surface of a manta ray (Stevens, pers. obs.). However, despite their locality, these large remoras where often still visible in the images because their heads or tails protruded from the edge of the host's body due to their favoured attachment positioning. Nonetheless, as the photo-ID images collected

in this study were primarily focused on the ventral surface of the rays, the presence of some *R. remora* were likely to have been missed. Future studies into associations between these two species should attempt to collect both ventral and dorsal imagery to address this methodological weakness.

Despite biological associations often being one of the first components of biodiversity to be altered by abiotic change, the associations between interacting species are often overlooked in regard to our changing world [5, 43, 86]. Disconcertingly, the climatic crisis and other anthropogenic threats in the Maldives are becoming increasingly apparent, with weakening monsoon winds, rising sea surface temperatures and levels, reef degradation, overfishing and habitat destruction all effecting the resilience of the ecosystems and the life they support [45, 87–89]. Changes in the Maldives environment determines the spatial and temporal variation in the presence and behaviour of manta rays [33, 34], which in turn drives ephemeral hitchhiker associations. This is an important consideration because the fitness benefits, and the degree of dependency between hitchhikers and manta rays, remain unknown, while declines in host species may alter hitchhiker populations [22, 58]. Research has already identified the potential for a reduction in the stability and pervasiveness of cleaner-client interactions under environmental change [90]. Thus, it begs the question of how stable hitchhiker associations will be under increasingly unstable environmental conditions [44, 91]. Therefore, an enhanced effort to document and understand these symbiotic interactions is critical.

## Conclusions

Manta rays provide a midwater habitat for a broad range of species that require the protection and sustenance these hosts afford [15, 22, 23, 68]. The current study identified a variation in the species of hitchhiker associated with *M. alfredi* and *M. birostris*, with *E. naucrates* and *R. remora* being the most common, respectively. Patterns of association in the presence of a range of hitchhiker species were identified, with spatiotemporal variation in the presence of manta rays acting as a driver for the occurrence of ephemeral hitchhiker associations. Of particular interest, near-term pregnant female *M. alfredi*, and *M. alfredi* at cleaning stations had the highest likelihood of an association with adult *E. naucrates*. Juvenile *E. naucrates* were more likely to be associated with juvenile *M. alfredi*, and a seasonal trend in *E. naucrates* host association was identified. Until now, these interactions have remained undocumented or briefly addressed in the literature.

Given the rapid pace at which anthropogenic activities are altering oceans worldwide, significant effort should be aimed at understanding these associations [5, 92]. The current study intends to serve as a basis for a deeper understanding of the symbiotic relationships and other associations which occur between manta rays and their hitchhikers, which in turn we hope will ultimately elucidate our knowledge of both the host and the hosted in a more ecologically meaningful way.

Further research of hitchhikers in different manta ray populations is warranted to evaluate whether the associations and structures found within the Maldives apply to other geographic locations, as well as understanding the drivers of the association more holistically. While it could be said that these hitchhikers are just along for the ride, they could also play a valuable role in the ecological understanding and conservation of such economically valuable and vulnerable species.

## Supporting information

**S1 Table. Summary of AICc relative goodness of fit metric values from the GLMM modelling procedure.** Includes all combinations of explanatory variales for each hitchhiker species

identified with *Mobula alfredi* and *M. birostris*.
(PDF)

## Acknowledgments

We thank the Manta Trust's resort and dive operator partners in the Maldives for their overall support with the study. We thank all Manta Trust staff, students and volunteers in the Maldives (past and present, especially Tam Sawers), as well as the marine biologists, water sports and dive teams throughout the country who contributed huge amounts of photo-ID data to this study. We also thank Peter McGregor. Finally, the authors would like to thank all the members of the public who submitted images to the Manta Trust for this study. We could not have undertaken this work without all your help.

## Author Contributions

**Conceptualization:** Aimee E. Nicholson-Jack, Guy M. W. Stevens.

**Data curation:** Guy M. W. Stevens.

**Formal analysis:** Aimee E. Nicholson-Jack, Joanna L. Harris.

**Funding acquisition:** Guy M. W. Stevens.

**Investigation:** Aimee E. Nicholson-Jack, Joanna L. Harris, Guy M. W. Stevens.

**Methodology:** Aimee E. Nicholson-Jack, Joanna L. Harris, Kirsty Ballard, Guy M. W. Stevens.

**Project administration:** Aimee E. Nicholson-Jack, Guy M. W. Stevens.

**Resources:** Guy M. W. Stevens.

**Supervision:** Katy M. E. Turner, Guy M. W. Stevens.

**Writing – original draft:** Aimee E. Nicholson-Jack, Joanna L. Harris.

**Writing – review & editing:** Aimee E. Nicholson-Jack, Joanna L. Harris, Kirsty Ballard, Katy M. E. Turner, Guy M. W. Stevens.

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
