## [Decision Letter · Decision Letter 0]

19 Mar 2021

PONE-D-21-04827

A hitchhiker guide to manta rays: patterns of association between *Mobula alfredi*, *M. birostris*, their symbionts, and other fishes in the Maldives

PLOS ONE

Dear Dr. Nicholson-Jack,

Thank you for submitting your manuscript to PLOS ONE. After careful consideration, we feel that it has merit but does not fully meet PLOS ONE’s publication criteria as it currently stands. Therefore, we invite you to submit a revised version of the manuscript that addresses the points raised during the review process.

Dear Dr. Nicholson-Jack,

Thank you for submitting your manuscript “A hitchhiker guide to manta rays: patterns of association between Mobula alfredi, M. birostris, their symbionts, and other fishes in the Maldives” to PLOS ONE. I have now received feedback from two experts in the field. As you can see below, they both enjoyed the paper but had a few comments. They provided constructive comments that will help you to improve your paper.

As a result, I would consider a resubmission addressing the minor comments made by these referees.

With kind regards,

Johann Mourier

We look forward to receiving your revised manuscript.

Kind regards,

Johann Mourier, Ph.D.

Academic Editor

PLOS ONE

Journal Requirements:

Additional Editor Comments (if provided):

Dear Dr. Nicholson-Jack,

Thank you for submitting your manuscript “A hitchhiker guide to manta rays: patterns of association between Mobula alfredi, M. birostris, their symbionts, and other fishes in the Maldives” to PLOS ONE. I have now received feedback from two experts in the field. As you can see below, they both enjoyed the paper but had a few comments. They provided constructive comments that will help you to improve your paper.

As a result, I would consider a resubmission addressing the minor comments made by these referees.

With kind regards,

Johann Mourier

Reviewers' comments:

Reviewer's Responses to Questions

**Comments to the Author**

1. Is the manuscript technically sound, and do the data support the conclusions?

Reviewer #1: Yes

Reviewer #2: Yes

2. Has the statistical analysis been performed appropriately and rigorously? 

Reviewer #1: I Don't Know

Reviewer #2: I Don't Know

3. Have the authors made all data underlying the findings in their manuscript fully available?

Reviewer #1: Yes

Reviewer #2: No

4. Is the manuscript presented in an intelligible fashion and written in standard English?

Reviewer #1: Yes

Reviewer #2: Yes

5. Review Comments to the Author

Reviewer #1: I only have four minor comments/remarks and one more general comment:

Lines 66-68: Consider rewriting. It’s not only megafauna but also deep-sea species, cryptic species etc.

Lines 96-98: I don’t find this a logic statement and/or necessary. I suggest deleting this sentence and to simply continue with “Here, we …”

Line 127: Adults/juveniles needs a reference.

Line 143: (log10 + 1) Why is this here, what does this mean?

Lines 352-356: The interspecific relationship between echeneids and their hosts remains difficult to classify, is complex, and likely not a simplistic symbiotic relationship; see e.g. Brunnschweiler et al. 2020 “The costs of cohabiting: the case of sharksuckers and their hosts at shark provisioning sites”. I suggest the authors discuss their results a bit more in this context, i.e. how do their (interesting!) findings add to our knowledge/understanding of the costs and benefits of the association for both organisms. Another paper, Norman et al. 2021 “Three-way symbiotic relationships in whale sharks”, published just a few days ago, might also be of interest to the authors. Did they, e.g. observe Remora remora feeding on plankton like Norman et al. 2021 did for R. remora associated with feeding whale sharks in the Maldives?

Reviewer #2: This manuscript describes a novel and interesting study into associations between manta rays and hitchhiker species in the Maldives archipelago. It is generally well-written and the authors draw on a large and valuable dataset to assess patterns of hitchhiker presence/abundance with their hosts, presenting an interesting angle that is yet to be explored extensively for these species.

The authors use a combination of Welch’s ANOVA and post-hoc tests and GLMMs to assess patterns of hitchhiker fish association with their manta ray hosts. Given the overall findings of the two analyses are the same for M. alfredi and similar for M. birostris, I would like to see an explanation of why it was necessary to include both analyses in the manuscript. Alternatively, the authors may consider including only one of these, which would make the manuscript more concise and streamlined. Regarding the GLMMs, there is no mention of how the assumptions of normality were assessed, or whether the authors tested for correlation between predictor variables? These are important elements that should be included.

Consideration of the influence of host body size on the presence/abundance/life-stage of hitchhikers appears to be missing from the manuscript. Body size is an obvious potential factor that may influence these associations, as increased surface area theoretically provides more space/protection for hitchhikers. Please see my specific comments below regarding sexual dimorphism and adult vs juvenile associations. Ideally, it would be valuable to include estimated disc width of individual manta rays as a predictor variable in the GLMMs, if these data are available, otherwise discussion of the influence of host body size is warranted.

I believe this study would be a valuable addition to the literature. However, at this stage I recommend minor revision and would gladly review the revised manuscript given the above points are addressed, as well as the specific comments below.

Specific comments:

Title

I am a big fan of the title!

Abstract

Line 24. I suggest changing to “hitchhiker fish species” (or including ‘fish’ somewhere in this sentence) on initial mention here so it’s clear you’re not referring to parasites

Lines 30-31. I suggest including which statistical methods were used here eg. “using Generalized Linear Mixed Effects Models”.

Lines 37-40. Perhaps add a brief sentence outlining main findings for M. birostris here.

Introduction

Lines 59-60. “…rather than the species themselves” I see what you’re saying here but without the species themselves there would be no interaction. I suggest re-wording to something like “rather than each species in isolation” or even “Such inter-specific interactions determine…” (here and in abstract)

Line 70. M. birostris has a circumglobal distribution, while M. alfredi has a semi-circumglobal distribution

Line 77. Do you mean “supports globally significant populations”? A single population of both species suggests they interbreed with one-another.

Line 85. “and there only”, delete ‘there’?

Line 88-91. I suggest swapping the order of these sentences. First explain that re-sighting rates are low, which, combined with seasonality of sightings, suggests a transient population which may be visiting these atolls as part of a ‘migration’. Then go on to say that where they are travelling from/to remains unknown.

Line 92. It would be useful to define the term ‘hitchhiker’ in the context of this study here, upon first mention. This way the reader knows exactly what you mean.

Line 96-97. I suggest simplifying to “Considering the widespread effects of anthropogenic activities on marine habitats”

Line 99. I suggest changing “presence” to “associations” or “in the presence of hitchhikers with.”

Line 102. associations (change to plural)

Line 103. What is a generative process? Do you mean the drivers of the associations? Consider re-wording

Materials and Methods

Line 111-112. This is correct, but the basis for compiling photo-ID catalogues is that individuals can be distinguished from one-another, not just from an existing catalogue, as new individuals are added over time. I suggest re-wording here. Also include the term “photo-identification” in full here as it is the first mention, with the abbreviation in parentheses.

Line 119-120. Was species identification/sex/maturity status determined on the dive or from photos? Please include how you determined sex (presence/absence of claspers) and maturity status for each sex, or cite a study which includes more detail on this.

Line 122. What determined whether a fish was a hitchhiker? For example, compared to a cleaner fish or just another fish in the vicinity of the manta ray, it would be useful to outline this here.

Line 122-123. The presence/number/species of hitchhikers was recorded, was this done during the dive? This can be tricky when there are a lot of mantas in the water at the same time (>15), in these cases were hitchhiker number and species determined by the photos? Please clarify.

Line 126. Please clarify whether the photo-ID images were ventral images, dorsal images or both. If only ventral, how did you assess the number of hitchhikers on the dorsal surface (as some species tend to move around the body of the host)? Or did you only include individuals that had both dorsal and ventral images?

Line 126. If the count from the dive did not match the count from the photo, how was this resolved?

Line 128. How was size measured? Was it estimated? If so, was this based on comparison with other objects of known size (perhaps the manta ray). Was there a way to resolve discrepancy between different observers and size estimates? As this can be variable among different people, particularly when estimating measurements underwater.

Line 128-129. Was the entire photo/record of the individual manta ray removed from analysis, or just the unidentifiable hitchhiker?

Figure 2. If these were the images used for identification, how were the images themselves identified? I suggest including a citation for the reference material used (i.e. a fish ID book).

Line 143. Why were the data transformed (log10 +1)? Please outline briefly here.

Line 145. ‘mean daily number of hitchhikers’ - is this all species combined? Or seprated by species?

Line 146. Here and elsewhere (line 219), I suggest changing ‘manta sexual group’. You could say ‘compared among sex and pregnancy stage’ or similar.

Line 147. Add ‘season’ after Monsoon.

Line 153. Please add a citation for the R package here.

Line 155. Which were the most frequently observed species? Include them here.

Line 158. Change to ‘sightings’, plural.

Line 159-160. How many sites were removed, and which sites?

Line 168. How did you assess whether your variables fit the normality assumptions for a GLMM? Did you test for correlations between predictors? Please include this here.

Line 170-172. Is there a reason why estimated/measured disc width wasn’t assessed? Perhaps surface area may influence the number/size/species of hitchhikers.

Line 173. Outline which species were the most frequently observed with M. birostris.

Line 177. Do you mean pseudoreplication?

Line 178. Is there a reason why the results of this goodness-of-fit comparison are not reported?

Line 182-183. This sentence can be moved to the results section.

Results

Line 196-200 and 254-255. Given the large numbers of sightings and individuals, percentages would be useful to include here.

Line 203. Is this 10% of total sightings? Or sightings with hitchhikers? Please clarify.

Line 218, 237 and 335. What is meant by daily mean number? The total number of E. naucrates seen in a single day? Or is this by sighting/individual? Please clarify.

Line 217 and line 236. Consider re-wording the opening sentences of these paragraphs. “The number of E. naucrates recorded ranged from 1-24 individuals”, or similar.

Line 257: “The highest percentage of E. naucrates presence by atoll” Is this based on the total or mean per individual manta?

Line 261-262. “Echeneis naucrates were not observed with M. alfredi during any sightings which occurred at 25 sites” Please re-word this sentence, do you mean that there were 25 sites at which E. naucrates were not observed?

Line 276-277. This belongs in methods section.

Line 279-280. Reference Table S1 here. With regard to Table S1, it is useful to report delta AICc in model selection tables and I would advise doing so here.

The second highest ranking model has delta AICc of <2. According to Burnham and Anderson (2004), when the difference in AIC between two models (delta AICc) is < 2, then there is substantial support for the two models having approximately equal weight in the data, and this rule-of-thumb is commonly adopted in the model selection process. Please explain in the methods section how you treated models within delta AICc of <2.

Burnham KP, Anderson DR (2004) Multimodel inference: understanding AIC and BIC in model selection. Sociol Method Res 33:261-304

Line 306. According to Table S1 the highest-ranking model (AICc = 6192.70) contains site function and maturity status as predictors, is the text correct or the table?

Line 275-311. The findings for the GLMM are the same as those for the Welch’s ANOVA and post-hoc tests for M. alfredi. Please explain why it is necessary to use both analyses.

Line 344. See my comment above about delta AICc of <2 in Table S1.

Discussion

I suggest starting the discussion with a paragraph that clearly summarises your main findings. This will guide the structure of the rest of the discussion to follow.

Line 378. E. naucrates were more likely to be associated with females. Have you considered the influence of sexual dimorphism? i.e. females are typically larger in size than males.

It would be interesting to assess whether there was any relationship between body size (disc width; if these data are available) and hitchhiker presence/abundance? Given that adult E. naucrates were more likely to be associated with adult M. alfredi (and the same with juvenile hosts and hitchhikers), is there a chance this is related to surface area? i.e. Larger surface area = more protection for larger hitchhikers.

This theory could also be applied to smaller hitchhiker species being more commonly associated with the smaller of the manta ray species (M. alfredi/E. naucrates) and larger hitchhikers with the larger of the manta species (M. birostris/ R. remora).

Line 383. Have you considered the nature of manta ray behaviour at different sites? For example, cleaning is characterised by slow movements and hovering, while feeding is typically faster movements, more turns etc. Could this make it harder for the hitchhikers to catch up/stay attached to their host?

Line 417. How is predation risk reduced at cleaning stations?

Line 418-435. This paragraph jumps between a number of themes/hypotheses and it distracts from the major point. Please re-word, and potentially separate the themes, for clarity.

Line 475 & 516-519. The fact that sightings images were primarily focused on the ventral surface of rays should be made clearer in the results section. See my comments above.

Line 491. What do you mean by population structure here? Demographic structure i.e. the distribution of adults v juveniles? Please clarify.

Line 499. Avoid referring to tables and figures in the discussion section.

6. PLOS authors have the option to publish the peer review history of their article (what does this mean?). If published, this will include your full peer review and any attached files.

Reviewer #1: No

Reviewer #2: No

---

## [Author Response · Author response to Decision Letter 0]

8 May 2021

Please see 'Response to Reviewers' document.

---

## [Decision Letter · Decision Letter 1]

3 Jun 2021

PONE-D-21-04827R1

A hitchhiker guide to manta rays: patterns of association between *Mobula alfredi*, *M. birostris*, their symbionts, and other fishes in the Maldives

PLOS ONE

Dear Dr. Nicholson-Jack,

Thank you for submitting your manuscript to PLOS ONE. After careful consideration, we feel that it has merit but does not fully meet PLOS ONE’s publication criteria as it currently stands. Therefore, we invite you to submit a revised version of the manuscript that addresses the points raised during the review process.

Dear Aimee Erin Nicholson-Jack,

Thank you for addressing all comments from both reviewers. They found you did a great job and that your work can now be accepted for publication. However they just noted a few minor points that I invite you to address before I can officially accept the paper and pass it to the production process.

Kind regards

Johann

We look forward to receiving your revised manuscript.

Kind regards,

Johann Mourier, Ph.D.

Academic Editor

PLOS ONE

Journal Requirements:

Additional Editor Comments (if provided):

Dear Aimee Erin Nicholson-Jack,

Thank you for addressing all comments from both reviewers. They found you did a great job and that your work can now be accepted for publication. However they just noted a few minor points that I invite you to address before I can officially accept the paper and pass it to the production process.

Kind regards

Johann

Reviewers' comments:

Reviewer's Responses to Questions

**Comments to the Author**

1. If the authors have adequately addressed your comments raised in a previous round of review and you feel that this manuscript is now acceptable for publication, you may indicate that here to bypass the “Comments to the Author” section, enter your conflict of interest statement in the “Confidential to Editor” section, and submit your "Accept" recommendation.

Reviewer #1: (No Response)

Reviewer #2: All comments have been addressed

2. Is the manuscript technically sound, and do the data support the conclusions?

Reviewer #1: Yes

Reviewer #2: Yes

3. Has the statistical analysis been performed appropriately and rigorously? 

Reviewer #1: Yes

Reviewer #2: Yes

4. Have the authors made all data underlying the findings in their manuscript fully available?

Reviewer #1: Yes

Reviewer #2: Yes

5. Is the manuscript presented in an intelligible fashion and written in standard English?

Reviewer #1: Yes

Reviewer #2: (No Response)

6. Review Comments to the Author

Reviewer #1: Thank you for addressing all my comments; I recommend to accept the ms for publication. However, there is one small thing I suggest to still consider: you have explained why you transformed data (log10+1). I would delete "(log10+1)" on lines 152 and 226. It's sufficient in my opinion to write "A (10 + 1) transformation was used for better visualisation of the data." (lines 227/228).

Reviewer #2: The authors have taken the time to carefully address the concerns raised by both reviewers and have done a thorough job revising the manuscript in response to the previous comments. The revised version is a much-improved paper and it is my opinion that the manuscript is now acceptable for publication in PLOS ONE.

Please also see a few minor comments to be addressed below:

Fig 1 is missing.

Line 152. ‘(log10+1)’ please expand on this as you have done in other parts of the manuscript.

Line 294, 307, 318, 325: The highest ranked model will always have delta AICc of 0, so no need to report it. Reporting on the AICc weight would be more useful if you would like to include a stat here, otherwise leave out.

Line 305, 315. It isn’t necessary to state if there were no models with delta AICc < 2.

7. PLOS authors have the option to publish the peer review history of their article (what does this mean?). If published, this will include your full peer review and any attached files.

Reviewer #1: **Yes: **Juerg M. Brunnschweiler

Reviewer #2: No

---

## [Author Response · Author response to Decision Letter 1]

6 Jun 2021

Please find our responses to the reviewers final comments at the beginning of the 'Response to Reviewers' document.

---

## [Editor Report · Decision Letter 2]

11 Jun 2021

A hitchhiker guide to manta rays: patterns of association between *Mobula alfredi*, *M. birostris*, their symbionts, and other fishes in the Maldives

PONE-D-21-04827R2

Dear Dr. Nicholson-Jack,

We’re pleased to inform you that your manuscript has been judged scientifically suitable for publication and will be formally accepted for publication once it meets all outstanding technical requirements.

Kind regards,

Johann Mourier, Ph.D.

Academic Editor

PLOS ONE
---

## [Editor Report · Acceptance letter]

21 Jun 2021

PONE-D-21-04827R2 

A hitchhiker guide to manta rays: patterns of association between *Mobula alfredi, M. birostris,* their symbionts, and other fishes in the Maldives 

Dear Dr. Nicholson-Jack:

I'm pleased to inform you that your manuscript has been deemed suitable for publication in PLOS ONE. Congratulations! Your manuscript is now with our production department. 

Kind regards, 

on behalf of

Dr. Johann Mourier 

Academic Editor

PLOS ONE